# Aging and Clonal Behavior of Hematopoietic Stem Cells

**DOI:** 10.3390/ijms23041948

**Published:** 2022-02-09

**Authors:** Masayuki Yamashita, Atsushi Iwama

**Affiliations:** Division of Stem Cell and Molecular Medicine, Center for Stem Cell Biology and Regenerative Medicine, The Institute of Medical Science, The University of Tokyo, 4-6-1, Shirokanedai Minato-ku, Tokyo 108-8639, Japan; 03aiwama@ims.u-tokyo.ac.jp

**Keywords:** hematopoietic stem cells, fate mapping, aging, clonal hematopoiesis

## Abstract

Hematopoietic stem cells (HSCs) are the only cell population that possesses both a self-renewing capacity and multipotency, and can give rise to all lineages of blood cells throughout an organism’s life. However, the self-renewal capacity of HSCs is not infinite, and cumulative evidence suggests that HSCs alter their function and become less active during organismal aging, leading ultimately to the disruption of hematopoietic homeostasis, such as anemia, perturbed immunity and increased propensity to hematological malignancies. Thus, understanding how HSCs alter their function during aging is a matter of critical importance to prevent or overcome these age-related changes in the blood system. Recent advances in clonal analysis have revealed the functional heterogeneity of murine HSC pools that is established upon development and skewed toward the clonal expansion of functionally poised HSCs during aging. In humans, next-generation sequencing has revealed age-related clonal hematopoiesis that originates from HSC subsets with acquired somatic mutations, and has highlighted it as a significant risk factor for hematological malignancies and cardiovascular diseases. In this review, we summarize the current fate-mapping strategies that are used to track and visualize HSC clonal behavior during development or after stress. We then review the age-related changes in HSCs that can be inherited by daughter cells and act as a cellular memory to form functionally distinct clones. Altogether, we link aging of the hematopoietic system to HSC clonal evolution and discuss how HSC clones with myeloid skewing and low regenerative potential can be expanded during aging.

## 1. Introduction

Hematopoietic stem cells (HSCs) are blood-forming stem cells that can clonally expand by self-renewing cell division, and that disseminate their clonal progeny by differentiating to all lineages of the blood and immune cells, such as leukocytes, erythrocytes and platelets [1]. Although some exceptions have been identified recently, such as tissue-resident macrophages [2,3] and innate-like B and T lymphocytes [4], HSCs are an important source for the vast majority of the blood and immune cells throughout an organism’s life, during steady-state hematopoiesis and hematopoietic regeneration after injury, thus being attractive targets for preventing or curing hematopoietic disorders. Even after the establishment and sophistication of efficient HSC purification methods, single-cell-based analyses have revealed a remarkable phenotypic and functional heterogeneity in the HSC pool [5]. Of note, recent advances in the clonal tracking methods, referred to as “fate mapping”, provided evidence that HSC heterogeneity can originate from individual clones that emerge during the developmental process [6,7]. Moreover, clonal analyses have revealed an age-related expansion of functionally skewed HSCs, which is associated with an imbalanced production between blood lineages, and reduced clonal diversity in hematopoiesis [6,8,9]. In this review, we will summarize the current understanding of HSC clonal behavior, since they emerge until the end of life, and discuss possible mechanisms that underlie functional diversity and age-dependent alteration in HSC clones.

## 2. HSC Development and Clonal Expansion

Cumulative evidence indicates that HSCs emerge mainly from the endothelial cells of the dorsal aorta in the aorta-gonad-mesonephros (AGM) region around E10.5 in mice and E30-42 in humans [10,11]. Upon budding into blood vessels, HSCs firstly migrate to the fetal liver, where they transiently amplify their numbers by self-renewing cell divisions. They then migrate to the bone marrow, where they continue to undergo self-renewing cell division until the end of the organism’s life. Of note, the majority of HSCs stop or slow down their cell cycle at around 4 weeks old in mice, and enter into quiescence, the G_0_ state of the cell cycle [12]. Thus, HSCs expand their clones from emerging during development, until they become quiescent in the adult bone marrow. How HSCs transition from the active to the quiescent state has not been clarified yet, but a recent single-cell RNA-sequencing (scRNA-seq) study suggests that the transition from fetal to adult HSCs occurs in a gradual, niche-independent, and stochastic manner and appears, at least in part, mediated by the type I interferon [13].

In the classical model, adult HSCs reside at the apex of a differentiation hierarchy and give rise to multipotent progenitors (MPPs), which do not possess a robust self-renewing capacity like HSCs, but still maintain the capacity to differentiate into all lineages of the hematopoietic cells. Along the course of differentiation, MPPs transiently amplify themselves and differentiate into more lineage-restricted progenitors, such as common myeloid progenitors (CMPs) and common lymphoid progenitors (CLPs), which undergo further amplification and commitments to become one of the mature blood cell types. Of note, recent transplantation analyses revealed remarkable functional heterogeneity within the phenotypically defined murine MPP subsets. Passegué and colleagues have shown that MPP2, MPP3 and MPP4 are already lineage-biased towards the erythro-megakaryocyte, myeloid and lymphoid lineages, respectively [14]. Cabezas-Wallscheid and colleagues have further demonstrated that MPP5 are functionally located between HSCs and the more committed MPP2-4, and are capable of stably contributing to both myeloid and lymphoid cell production upon transplantation [15]. Moreover, rather than the classical step-by-step decision making model depicted by the differentiation hierarchy, scRNA-seq studies demonstrated a continuum of the lineage commitment from HSCs to each mature cell type, with the majority of the decision making processes being initiated at the hematopoietic stem and multipotent progenitor cell (HSPC) level [16,17,18,19,20]. These findings have led to the development of a revised model of adult HSC differentiation (Figure 1), the details of which have been discussed elsewhere [21].

Quiescence is considered as an important mechanism to compensate for HSCs’ incomplete self-renewal ability, and for reserving as many functional HSCs as possible throughout life. This concept is supported by the fact that HSCs often lose their self-renewal ability if they are forced to undergo cell division by ex vivo cell culture [22] or in vivo serial transplantation [23], massive loss of blood cells by traumatic injury [24] and other insults to the bone marrow, such as myeloablative drugs [25]. In addition, by using a molecule that is diluted by half upon cell division, such as Bromodeoxyuridine (BrdU) and the histone H2B-GFP fusion protein, quiescent, label-retaining HSCs are shown to possess a more durable self-renewal capacity compared to actively cycling, non-labeled HSCs [23,26]. Interestingly, the label-retaining assay using H2B-GFP mice suggested that HSCs lose their self-renewal capacity after four cycles of self-renewing cell division [27]. This indicates that HSCs can remember their divisional history through certain molecular mechanisms, which restrict their self-renewal ability. The same group suggested in the following paper that a switch from EZH1 to EZH2, the two catalytic subunits of polycomb repressive complex 2 (PRC2), occurs during successive divisions and could explain the loss of self-renewal in HSCs [28]. Although this hypothesis appears intriguing, as discussed in a later section, there is an argument against the H2B-GFP-based estimation of divisional history [29,30], and alternative methods to precisely monitor HSC divisional history in vivo would be necessary for further investigation. In sum, upon emergence, fetal HSCs actively expand their clones by self-renewing cell division to make up a diverse pool of adult HSCs and constitute a spectrum of differentiated progeny. However, by the time HSCs are expanded enough to establish hematopoietic homeostasis, they are likely instructed to limit their self-renewal ability and enter quiescence in order to maximize their longevity and minimize the risk of malignant transformation.

## 3. HSC Functional Diversity and In Vivo Fate Mapping

Fate mapping is a method used to analyze where a certain cell or tissue comes from in the process of development and regeneration. Historically, random retroviral integration was used to mark HSCs and evaluate their behavior, by detecting the genomic integration sites after transplantation [31]. A more sophisticated version of this is to label HSCs ex vivo with retroviral or lentiviral barcoding libraries and transfer them to mice [32,33,34]. In humans, Biasco and colleagues have characterized the frequency, dynamics and output of HSPC subtypes after autologous transplantation, by tracking lentiviral integration sites in gene therapy-treated Wiskott-Aldrich syndrome patients [35]. However, these methods essentially require transplantation, and therefore cannot analyze the clonal composition of HSCs during development or aging. Instead, various methods have been devised to label HSC clones and evaluate their behavior in vivo, and some of the paradigms built upon transplantation-based experiments have been revised. Here, we will introduce recent advances in HSC clonal tracking strategy and discuss what has been drawn from such HSC clonal analysis in vivo.

### 3.1. Fluorescence-Based Tracking

Inducible fluorescent reporter mice have been a powerful tool for monitoring the spatiotemporal behavior of labeled cells in vivo, and numbers of HSC-specific reporters were created to track the fate of labeled HSCs (Table 1). For example, Begley and colleagues have created transgenic mice that express tamoxifen-inducible Cre under the control of a *Scl* 3′ enhancer, and demonstrated the multilineage contribution of embryonic HSPCs into adult hematopoiesis [36]. Rodewald and colleagues have generated knock-in mice where tamoxifen-inducible, codon-improved Cre was fused to two modified estrogen receptor binding domains, which were knocked into the first exon of the *Tie2* locus (*Tie2-MCM*). This produced a remarkably low contribution of labeled adult HSCs to downstream progeny [37]. In contrast, Reizis and colleagues have established transgenic mice using a bacterial artificial chromosome (BAC) transgene, in which tamoxifen-inducible Cre^ERT2^ was inserted into the first exon of the *Pdzk1ip* locus. This demonstrated a multilineage contribution of adult HSCs to steady-state hematopoiesis, except B-1a cells and tissue-resident macrophages [38]. Furthermore, knock-in mice have been developed by Rossi and colleagues via knocking Cre^ERT2^ into the first exon of the *Fgd5* locus [39], with which the active contribution of HSCs to the steady-state hematopoiesis and the reduction in their multipotency during aging, have been demonstrated [40]. The active hematopoiesis by steady-state phenotypic HSCs has been further confirmed by Nakada and colleagues using other HSC-specific *KRT18-Cre^ERT2^* mice [41]. Nonetheless, these single-color reporter mice cannot distinguish individual HSC clones and therefore are not suitable for evaluating clonal architecture. To circumvent this problem, a stochastic multicolor Cre recombinase reporter has been developed to randomly label stem cells with different colors and track their in vivo clonal behavior. This system is quite useful for visualizing the clonal behavior of tissue-specific stem and progenitor cells in the solid organs, such as the brain and intestine [42], and similar attempts have been made to evaluate HSC clonal behavior after transplantation or during aging [9,43]. Interestingly, by combining genetic color-coding of zebrafish mesoderm with the CRISPR-Cas9-mediated mutagenesis of several clonal hematopoiesis-related genes, Zon and colleagues have recently demonstrated the clonal expansion of *Asxl1*-mutant HSCs, with the support of an inflammatory environment [44]. However, since hematopoietic tissues are not compartmentalized like the intestine and the ability to distinguish individual clones is hampered due to spectrum overlapping, the visualization of clonal architecture of hematopoiesis by multicolor fluorescence remains relatively challenging.

### 3.2. Genome DNA Barcoding

Recently, in vivo cell barcoding systems with higher resolution have been developed by multiple groups and utilized to evaluate HSC clonal behavior. For instance, Camargo and colleagues have created a mouse system where a subset of murine HSCs and hematopoietic progenitors are labeled in vivo by the doxycycline-dependent transient induction of Hyperactive Sleeping Beauty (HSB) transposases and the subsequent mobilization of a tagged DNA transposon to elsewhere [47]. As the insertion of a transposon occurs almost randomly, the insertion site can be used as a unique genetic barcode. Of interest, in non-transplantation settings, the majority of the barcodes detected in peripheral leukocytes are also present in multipotent or oligo-potent progenitors, but cannot be detected in HSCs even after 10 months post-HSB-based barcoding. In contrast, a year after the transplantation of barcoded HSCs, many of the barcodes observed in peripheral leukocytes are present in both HSCs and their progeny. They have further demonstrated that while the functional heterogeneity of MPP subsets in unperturbed bone marrow is largely in line with that observed in transplantation settings, with MPP2 showing biased output toward megakaryocyte progenitors, MPP3 toward granulocyte and monocyte progenitors, and MPP4 toward lymphoid and multilineage output, a significant portion of HSCs show megakaryocyte-restricted output in unperturbed settings, but multilineage output upon transplantation [51]. Although there is still a debate, in particular with regard to the labeling efficiency of HSCs in this system [38], these observations point out that while HSCs actively contribute to hematopoiesis upon the transplantation that inevitably accompanies various hematopoietic stresses, non-self-renewing multipotent progenitors could be a main source of native, unperturbed hematopoiesis where such stress does not exist, arguing against the dogma of “bona fide” HSCs that had long been defined by transplantation experiments.

Another in vivo barcoding system has been developed by Rodewald and colleagues and applied to HSC clonal tracking [7]. In their system, a *Polylox* DNA cassette that consists of ten *loxP* sites in alternating orientations spaced 178 base pairs apart, was integrated into a mouse genome, and Cre recombinase-mediated excision and inversion were induced to generate diverse recombined *Polylox* sequences that can be used as cellular barcodes. Consistent with the observation with the HSB-based method, the *Polylox* barcoding of HSCs in adulthood by *Tie2-MCM* has shown a small output of HSCs to peripheral leukocytes. Of note, the selective labeling of fetal HSCs at E9.5 has revealed differential clonal behavior within HSC clones during development, with some clones estimated to expand to several hundreds of cells until they become quiescent in the adult bone marrow, while others are maintained with minimal expansion. Furthermore, some clones showed a balanced contribution to myeloid and lymphoid lineages, while others showed a skewed differentiation to myeloid cells. Thus, it is conceivable that during the developmental process in the fetus, individual HSC clones with different lineage potentials expand asymmetrically, leading to various large and small HSC clones with functional heterogeneity in the adult HSC pool (Figure 2a–c), which is further supported by a recent scRNA-seq analysis [13].

### 3.3. mRNA Barcoding

While genome barcoding is a powerful tool for tracking the fate of labeled stem cell clones in vivo, this cannot provide a mechanistic insight into how cell fate is determined for each clone. To circumvent this point, the barcode has been designed to be transcribed as a part of mRNA, so that transcriptome and barcoding can simultaneously be evaluated in a cell by scRNA-seq. CRISPR array repair lineage tracing (CARLIN) and *PolyloxExpress* are such examples of mRNA-based single-cell barcoding [48,49]. Interestingly, by overlaying clonal information on a transcriptome-based differentiation trajectory, these studies have independently identified differentiation-inactive, childless HSCs during unperturbed hematopoiesis, which were defined as HSC clones that do not share barcodes with MPPs or other downstream hematopoietic progeny. These inactive HSCs were still observed upon myeloablation by 5-fluorouracil (5-FU) treatment, where virtually all HSCs enter the cell cycle [48], indicating that the childless output is a property distinct from quiescence. Of note, the childless output was associated with gene signature. represented with major histocompatibility (MHC) class II and its related gene CD74, but not with cell cycle-related genes. As these signature genes are also upregulated in aged HSCs [53], it would be interesting to determine whether such inactive HSCs accumulate in the bone marrow of aged organisms.

### 3.4. Mitochondrial Barcodes

The above systems have a great capacity to track lineages out of HSCs in model organisms, but they are not feasible in human hematopoiesis. Traditional lineage-tracing studies of human hematopoiesis require the transplantation of lentiviral vector-transduced HSPCs for gene therapy [35,54]. Even though there are a few lineage-tracing studies in intact human tissues using naturally occurring nuclear somatic mutations [55,56], the detection of such mutations has to rely on whole genome sequencing, which is difficult to combine with the single-cell technique, and which precludes comprehensive clonal analysis of native hematopoiesis in humans. Sankaran and colleagues have focused on mitochondrial DNA (mtDNA) that acquires mutations more frequently than nuclear DNA. They revealed that mtDNA mutations in individual adult human HSCs are diverse enough to be distinguished from each other, and can be utilized as cellular barcodes that work as efficiently as lentiviral barcodes [50]. As single-cell mtDNA sequences can reliably be assessed by single-cell assays for transposase-accessible chromatin using sequencing (scATAC-seq) and scRNA-seq, this approach holds promise for retrospectively tracking the clonal behavior of human HSCs in the setting of native hematopoiesis, leukemogenesis and aging.

## 4. HSC Clonal Behavior during Aging

Aging in the hematopoietic system leads to disrupted hematopoietic homeostasis, such as anemia, a deregulated immune system and an increased propensity to hematological malignancies in aged individuals. Most, if not all, of these age-dependent changes in hematopoiesis are closely associated with, and often attributed to, age-related alterations in HSCs. Such alterations include an increase in the number of phenotypic HSCs, a reduction in their self-renewal and regeneration potential, an impaired homing to the bone marrow, a skewed differentiation to myeloid and megakaryocytic lineage cells and reduced production of lymphoid progenitor cells [53,57,58,59]. Of note, a preceding study on X-chromosomal inactivation has revealed an age-dependent progressive increase in the clonality of blood leukocytes [60]. The clonal analysis of HSCs with single fluorescent color demonstrated an age-related differentiation block between HSCs and MPPs [40]. Furthermore, Nakauchi and colleagues have analyzed the clonal behavior of transplanted single HSCs of aged mice, and revealed a marked alteration in their clonal composition [8]. Even though the majority of aged HSCs are either myeloid-biased clones with less or no self-renewing capacity, a small portion of them maintain long-term repopulating potential with balanced myeloid and lymphoid output. Moreover, there are some latent HSC clones only observed in aged mice that undergo limited differentiation upon first transplantation, but show multilineage output after secondary transplantation. These results point to the age-related expansion of HSC clones with relative resistance to differentiation-instructing stimuli, and although these latent HSCs are essentially defined by transplantation experiments, it is possible that they may also be childless in the native hematopoiesis of aged organisms (Figure 2d). The heterogeneity of aged HSCs is also demonstrated in terms of transcriptome [53,61,62,63], autophagic activity [64], mitochondrial membrane potential [65] and cell size [66]. Such remarkable heterogeneity in the aged HSC pool indicates that individual HSCs are differentially affected in the course of aging, perhaps depending on their behavioral histories (Figure 3a). Below we describe age-dependent cell-intrinsic and -extrinsic changes and discuss how these alterations could affect the clonal behavior of individual HSCs.

### 4.1. Somatic Mutation

Most of the aged HSC phenotypes are largely maintained after transplantation into young recipient mice [58]. Thus, age-dependent functional decline could be attributed to intrinsic changes that are accumulated in HSCs during aging. Human HSCs are estimated to accrue approximately 10–14 base substitutions per year [67], raising the possibility for mutation accumulation to serve as age-dependent cellular memory (Figure 3b). Indeed, multiple groups have performed large-scale DNA sequencing studies with human blood samples and have independently revealed the existence of somatic mutations in peripheral leukocytes without hematological malignancies [68,69,70]. This outgrowth of mutant clones, termed “clonal hematopoiesis”, often accompanies mutations of genes involved in HSC competitive fitness and differentiation, such as *DNMT3A* [71,72], *TET*2 [73,74], *ASXL1* [75], *JAK2* [76,77], *TP53* [78,79], *PPM1D* [80,81], and thus believed to be derived from the clonal expansion of mutant HSCs. Of note, the prevalence of clonal hematopoiesis increases with age, becoming apparent around 40 years of age and reaching 10–20% over age 70, indicating that clonal hematopoiesis emerges as a result of age-dependent mechanisms [69,70]. Thus, it is conceivable that human HSCs acquire specific mutations and expand clonally during aging, leading to clonal hematopoiesis where a portion of mutant HSC subsets give rise to mutant mature hematopoietic cells. Of note, clonal hematopoiesis is shown to be an independent risk factor for atherosclerotic cardiovascular diseases, and the risk of coronary heart disease is estimated to increase ~two-fold with *DNMT3A*, *TET2* and *ASXL1* mutations, and ~10-fold with the *JAK2V617F* mutation [82]. Whether and how clonal hematopoiesis should be managed is a topic of active discussion [83].

However, aged HSC phenotypes significantly overlap between mice and humans [84], even though mice have a much shorter lifetime (~2 years) compared to humans (~70 years) and seemingly accumulate fewer mutations until the end of life [9]. Moreover, the mutations that frequently occur in human clonal hematopoiesis do not seem to occur in mouse HSCs during aging [9]. Although more comprehensive work remains to be done on the mutational landscape of aged HSCs, including non-coding as well as coding regions of the genome, the considerable similarity of the aged HSC phenotype between mice and humans rather suggests the existence of non-genetic mechanisms underlying HSC aging.

### 4.2. Epigenetic Memory

Epigenetic modifications regulate gene expression without changes in the DNA sequence, and are important determinants of cell fate. Some of the epigenetic modifications, including DNA methylation and post-translational histone modifications, such as PRC2-mediated histone H3 lysine 27 trimethylation (H3K27me3) and heterochromatin-associated lysine 9 trimethylation (H3K9me3), are known to be mitotically inheritable, and can be retained in daughter cells upon cell division, and clonally propagated in the progeny [85]. HSC fate decision is also guided by the epigenetic status, and the deregulation of epigenetic regulators significantly alters HSC function [71,73,86,87].

Various age-dependent epigenetic changes have also been identified, and the genes involved in epigenetic regulations are frequently mutated in human clonal hematopoiesis. For example, DNMT3A and TET2 are involved in DNA methylation and demethylation, respectively (Figure 3b). Rossi and colleagues have demonstrated the proliferation-dependent DNA hypermethylation of PRC2 target genes in aged murine HSCs [25]. Such age-dependent changes in DNA methylation are refractory to exposure to the young bone marrow environment and seem to underlie many of the HSC functional changes, including reduced engraftment and lymphopoietic potential [88,89]. DNA methylation status is closely linked to histone marks, and Goodell and colleagues have indeed revealed a PRC2-mediated H3K27me3 increase in both the length of coverage, by 29%, and in average signal intensity at the transcription start sites, by ~50%, despite similar H3K27me3 peak counts in aged murine HSCs [90]. On the other hand, Figueroa and colleagues have demonstrated that the signal intensity of H3K27me3 peaks decreases with age in a human HSC-enriched population [91], highlighting the complexity of age-related changes in PRC2 activity. The expression of *Ezh1* may increase with age [90,92], albeit this observation appears not always to be reproduced [93]. Although the function of EZH1 and EZH2 seems largely overlapping, EZH1 has non-redundant, ontogeny-specific, and dose-dependent roles, such as inhibition of HSC emergence before definitive hematopoiesis via repression of HSC signature genes, and the prevention of adult HSC depletion by *Cdkn2a* repression [92,94]. While EZH2 appears largely dispensable for HSC function [94,95], the pharmacological inhibition of EZH2 catalytic activity over EZH1 seems to prevent cell division-dependent loss of HSC self-renewal, perhaps by dominating EZH1 function over EZH2 [28]. Thus, the age-associated overexpression of EZH1 and cell division-coupled PRC2 switching from EZH1 to EZH2 may underlie the phenotypic expansion and functional decline of HSCs during aging (Figure 3c). 

On the other hand, Goodhardt and colleagues have reported a global reduction of heterochromatin-associated repressive histone mark H3K9me3 and a reduced expression of the histone methyltransferase SUV39H1 in aged murine and human HSCs [96], which correlates with the derepression of the transposable elements (TEs), such as LINEs, SINEs and LTRs [96]. As recognition of the derepressed TE expression by MDA5, it has recently been suggested that an intracellular receptor for double-strand RNAs is critical for 5-FU-induced HSC cell cycle entry, as well as for the type I interferon response [97]. A similar mechanism could be involved in age-related HSC alternation (Figure 3c). 

DNA methylation and histone modification can regulate the expression of not only protein-coding RNAs but also non-coding RNAs, such as long non-coding RNAs (lncRNAs) and microRNAs (miRNAs). These non-coding RNAs play critical roles in many features of adult HSCs [98], and emerging evidence suggests the roles of such non-coding RNAs in HSC aging. For example, Baltimore and colleagues have shown that miR-132 is overexpressed in aged murine HSCs and is involved in the protection of HSC survival and balanced lineage output by directly targeting FOXO3A and promoting autophagy [99]. In addition, Karsan and colleagues have indicated that the loss of miR-146a, which mainly targets TRAF6 and IRAK1 and thus can act as a rheostat of the Toll-like receptor (TLR) and the NF-κB pathway, may drive HSC aging by increasing immune cell-derived inflammatory signals and HSC sensitivity to inflammatory cytokines, such as IL-6 and TNF-α [100]. Moreover, Goodell and colleagues have identified 29 lncRNAs that are enriched in HSCs compared to mature hematopoietic cells and which are differentially expressed between young and aged HSCs, albeit their functions remain uninvestigated [101].

Taken together, epigenetic memory is likely a mechanism that can mediate the age-dependent conversion of HSCs to functionally altered clones. Obviously, further investigation is needed to clarify how these age-related epigenetic changes coordinate with each other to mediate age-related HSC functional decline. In addition, since recent studies using scATAC-seq revealed heterogeneity in chromatin accessibility of fetal HSCs that is closely linked to their lineage priming [102], it would be interesting to see how such epigenetic heterogeneity is affected during aging.

### 4.3. Mitochondrial Inheritance

Mitochondria are now recognized as not only a cellular “power plant” but also a central hub of cellular biochemistry and stress response. Aside from energy metabolism and adenosine triphosphate (ATP) production via the tricarboxylic acid (TCA) cycle and oxidative phosphorylation (OXPHOS), mitochondria are the site of many other metabolic processes, such as those of amino acids, lipids and nucleotides [103]. Mitochondria have their own DNA (mtDNA) and other bioactive molecules inside an inner membrane and are physically associated with the endoplasmic reticulum (ER) and the lysosome, serving as an important signaling hub for various stress response pathways, such as apoptosis and the unfolded protein response (UPR). Mitochondria dynamically respond to cellular stress or mitotic signals by changing their distribution and network through the autonomous membrane remodeling, termed fission and fusion, or by actively removing damaged components via autophagic degradation, termed mitophagy [104]. Despite such quality control mechanisms, the functionality of mitochondria can be impaired by various stress. Mitochondria can be damaged due to the activation of pore-forming proteins, such as BAX and BAK, with no or little activation of caspases, leading to incomplete or minority mitochondrial outer membrane permeabilization (MOMP), without inducing cell death [105]. Indeed, Kile and colleagues have shown the role of BAX/BAK-mediated MOMP in releasing mtDNA and the type-I IFN-mediated expansion of dysfunctional HSCs in a caspase-inactivated setting [106]. Damaged mitochondria in HSCs can be inherited by daughter cells upon cell division and propagated in the hematopoietic system, leading to the impairment of hematopoietic integrity.

Emerging evidence indicates that mitochondrial quantity and quality are a critical regulator of HSC regenerative capacity and lymphopoietic potential. Historically, HSCs have been shown to contain a low abundance of mitochondria using mitochondria-specific dyes, such as MitoTracker Green [107,108]. However, staining with such cell permeable dyes can be affected by xenobiotic efflux pumps, such as the ATP-binding cassette (ABC) transporters, which are preferentially expressed in HSCs among hematopoietic cells. Consequently, a combination of mitochondrial dyes with an efflux pump inhibitor verapamil or dye-independent methods, such as MitoDendra2-GFP fusion reporter system, has rather revealed high mitochondria content in HSCs compared to committed progenitors [109,110], as well as quiescent vs. active HSCs [111]. Along a similar line, mitochondrial membrane potential (MMP), a key determinant of mitochondrial activity typically measured by cationic dyes, such as tetramethyl rhodamine ethyl or methyl ester (TMRE or TMRM, respectively) and DilC1(5), was initially estimated to be low [107,108,112,113], but turned out to be high in HSCs with verapamil treatment [65,110] (Figure 3d). Thus, despite the heavy reliance to glycolysis compared to OXPHOS for energy metabolism [114], mitochondrial content and membrane potential seem to be kept high in quiescent HSCs. Mitochondrial fusion activity via PRDM16-MFN2 is shown to be critical for maintaining the lymphoid potential of HSCs, suggesting that the age-dependent decline in HSC clones with robust lymphoid potential may be linked with a defect in mitochondrial fusion [115]. 

Interestingly, recent evidence suggests that mitochondria, as well as lysosomes, can be asymmetrically segregated to daughter cells upon HSC division. Ghaffari and colleagues showed that quiescent HSCs contain a punctate mitochondrial network with abundant lysosomes and suggested that mitochondria are actively degraded by lysosomes to maintain their quiescence [116]. Indeed, Passegué and colleagues have demonstrated that ATG12-mediated autophagic activity is essential to maintain engraftment and the lymphopoietic capacity of HSCs [64]. Furthermore, Schröder and colleagues have performed in vitro continuous single-cell imaging and tracking to reveal that HSCs can asymmetrically inherit lysosomes and mitophagosomes during cell division, and daughter cells with a low number of lysosomes tend to undergo metabolic and translational activation and commitment, further highlighting the role of mitophagy in the maintenance of HSC quiescence and function [117]. Filippi and colleagues further indicated that transplantation- and 5-FU-induced replicative stress causes irreversible changes in mitochondrial morphology and function at the HSC level, including a reduction of MMP, which can be inherited by daughter HSCs and maintained even after they re-enter quiescence, likely due to the decline in DRP1-mediated mitochondrial fission activity [118]. Therefore, it is possible that as a result of exposure to cellular stress that injures their mitochondria, HSC clones with damaged mitochondria may accumulate in the course of organismal aging.

In support of this hypothesis, HSCs with low MMP and fragmented, dysfunctional mitochondria are accumulated in aged mice, leaving only 10–33% of remaining HSCs relatively functional [64,65,119]. Of note, recent findings by Enver and colleagues suggested that age-dependent alteration in mitochondrial function may play a causative role in reducing the regenerative and lymphopoietic potential of HSCs, as the pharmacological potentiation of mitochondria using a mitochondrial-targeted coenzyme-Q10 mitoquinol (Mito-Q), ameliorated the decreased engraftment capacity and myeloid-biased differentiation of aged HSCs. Along a similar line, in vivo supplementation of nicotinamide adenine dinucleotide (NAD^+^) precursor, nicotinamide riboside (NR), was shown to improve the mitochondrial function and lymphoid potential of aged HSCs [120,121]. Thus, it is tempting to speculate that mitochondria can behave as a cellular memory for HSC divisional history and stress response. HSCs with higher levels of mitochondrial damage may have reduced regenerative and lymphopoietic potential, and the aging process may favor the accumulation of such HSC clones. Further studies are absolutely needed to answer many remaining questions as to how mitochondrial content and turnover are regulated in HSCs, what causes mitochondrial damage in HSCs during aging, why HSCs keep damaged mitochondria, and how mitochondria damage is translated to age-related changes in HSC function.

### 4.4. Roles of Niche in Clonal Selection

Aging also involves structural and functional changes in the bone marrow microenvironment, which can contribute to the selection of specific HSC clones with lymphoid potential. Indeed, Adams and colleagues have revealed the age-dependent loss of arteriolar endothelial cells and periarteriolar PDGFRβ^+^/NG2^+^ mesenchymal stromal cells (MSCs) in bone marrow, which results in reduced stem cell factor (SCF) production and which can be reverted by endothelial Notch activation [122]. Endothelial cells also express various Notch ligands, among which DLL4 was shown to be vital for preventing myeloid skewing of HSCs and the loss of common lymphoid progenitors (CLPs). Furthermore, Trowbridge and colleagues have shown that insulin-like growth factor 1 (IGF1), which is locally produced by Nestin^+^ bone marrow MSCs to selectively stimulate expansion of HSCs with robust lymphoid potential, begins to decline at middle age [119]. Of note, Morrison and colleagues have demonstrated that leptin-receptor (LEPR)- and osteolectin-expressing periarteriolar osteoprogenitors are critical for maintaining CLPs, but no other hematopoietic progenitors, and are depleted with age, suggesting that the age-dependent impairment of lymphoid production also occurs at the CLP stage [123]. Interestingly, they further indicated that these CLP-specific niche cells can sense mechanical stimulation via mechanosensitive PIEZO1 channels and are depleted when mechanical stimulation is reduced, highlighting the importance of exercise and mechanical loading in lymphoid cell production and acquired immunity. On the other hand, age-related neuropathy in bone marrow is also implicated in biased hematopoiesis. Frenette and colleagues have shown the age-dependent remodeling of sympathetic nerves and the degeneration of niche-associated β3 adrenergic neurons in bone marrow, which underlies the loss of HSCs with lymphoid potential in aged mice [124]. Mendez-Ferrer and colleagues further revealed that an age-dependent increase in the innervation of β2 adrenergic neurons promotes megakaryopoiesis through IL-6 production, which is another hallmark of skewed hematopoiesis in aged organisms [125].

Chronic sterile inflammation underlies the age-related deregulation of tissue integrity, and HSCs express various receptors for pro-inflammatory cytokines that enable them to rapidly change their fate in response to inflammatory stimuli. Indeed, proinflammatory cytokines, such as IL-1, TNF-α, and RANTES, are increased in the bone marrow of aged mice and implicated in the age-dependent myeloid skewing of hematopoiesis [126,127]. IL-1, TNF-α, and IFN-γ are also implicated in the selective expansion of mutant HSC clones over wild-type counterparts, suggesting the role of the inflammatory environment in HSC clonal selection [128,129,130,131,132]. For example, IL-1 is shown to promote precocious HSC differentiation to myeloid lineage cells via the activation of the NF-κB-PU.1 axis [133], and to facilitate the expansion of *C/EBPα*-mutant HSPC clones [130]. Moreover, a recent mouse study has revealed that TNF-α can act as a pro-survival and pro-regenerative factor specific for HSCs and MPP2-3 by inducing a specific set of canonical NF-κB/p65 target genes that promote HSC survival, inhibit T cell activity via the PD-1 pathway, and activate emergency myelopoiesis [134]. Of note, such HSC-specific TNF-α signature genes are also indicated to be upregulated in aged and malignant HSCs, and TNF-α is indeed shown to promote the expansion of *Tet2*-deficient, *Fancc*-deficient and *JAK2V617F*-harboring mutant HSPCs [128,129,132]. Interestingly, Aifantis and colleagues have recently shown that the E3 ubiquitin ligase Speckle-type BTB–POZ protein (SPOP) restrains the IL-1-dependent inflammatory activation of HSCs through ubiquitylation and the proteasomal degradation of MYD88, and the perturbation of such post-translational modification causes a persistent inflammatory response in HSCs [135]. Thus, it is possible that increased levels of inflammatory cytokines and impaired resolution of inflammatory response could together contribute to the age-dependent expansion of myeloid-biased HSPCs and mutant HSPC clones.

The source of such inflammation remains obscure, but the role of pattern recognition receptor (PRR) ligands, such as the pathogen-associated molecular pattern (PAMPs) and damage-associated molecular patterns (DAMPs), has been proposed. These PRR ligands are typically sensed by immune cells in order to initiate inflammation, but can directly be recognized by HSCs as well, since HSCs are also shown to express various PRRs, such as TLRs and nod-like receptors (NLRs) [136,137]. For example, a modest increase in circulating PAMPs has recently been reported in mice, which may underlie the overproduction of IL-1α/β and contribute to myeloid-biased hematopoiesis [138]. Moreover, hematopoietic cells deficient for *Tet2* were shown to cause gut barrier malfunction and secondary bacterial translocation, leading to IL-6 overproduction, which supports malignant transformation [139]. On the other hand, circulating DAMPs are also known to be increased in an age-dependent manner, and S100A8/9 are indeed indicated to be released from MSCs in bone marrow and to induce genotoxic stress in HSCs. Notably, emerging evidence indicates that HSCs play a central role in trained immunity, a form of innate immune memory that is important for efficient pathogen clearance after a secondary challenge [140,141,142]. Mechanistically, mouse studies suggest that infection history can be inscribed at the HSC level via epigenetic or metabolic reprogramming, such that HSCs can rapidly provide innate immune cells upon secondary infection [142,143]. Furthermore, Baltimore and colleagues have performed scRNA-seq of HSPCs isolated from mice stimulated with lipopolysaccharide (LPS), a TLR4 ligand that is present on the outer membrane of most Gram-negative bacteria, and have identified CD61-high HSCs as myeloid-biased subsets that are highly responsive to LPS and massively expanded in aged mice [62]. Such myeloid-biased HSCs can expand themselves, but also be generated from CD61-low HSCs with balanced lineage output, perhaps due to an alteration in enhancer methylation status [43], indicating that both clonal selection and cell-intrinsic conversion underlie the age-associated expansion of myeloid-biased HSCs. Therefore, it is tempting to speculate that once an HSC is exposed to such PRR ligands, that experience is memorized by the HSC, perhaps through epigenetic modifications or mitochondrial alterations, and such “memory HSCs” accumulate in the bone marrow during aging.

## 5. Conclusions and Perspectives

The recent progress in single-cell analysis and fate mapping tools allows researchers to track the behavioral history of individual HSC clones in vivo. HSCs with various self-renewal and differentiation capacities appear at the very early stage of development and clonally expand or shrink in response to various cell-intrinsic and environmental changes, leading to the establishment of functionally heterogeneous HSC pools in bone marrow. The aging process reduces the number of active clones contributing to mature hematopoiesis, and further modifies the functional diversity in the HSC pool, perhaps by favoring the clonal expansion of myeloid-biased or childless HSCs, leading to the deterioration of the hematopoietic system. Cumulative evidence indicates that age-related clonal hematopoiesis driven by mutant HSCs, serves as a non-negligible risk factor for various life-threatening disorders, such as hematological malignancy and cardiovascular diseases. Deconvoluting clonal behavior of HSCs during aging will provide clues for understanding the nature of HSC aging and for preventing diseases caused by age-related clonal hematopoiesis.

## Figures and Tables

**Figure 1 ijms-23-01948-f001:**
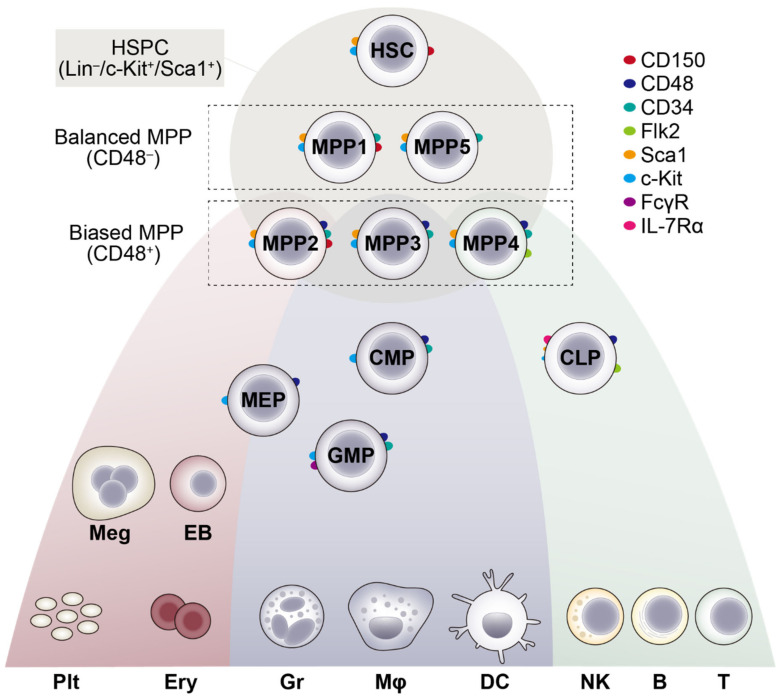
A revised model of adult HSC differentiation. In both humans and mice, recent scRNA-seq analyses predicted a revised model of adult HSC differentiation, where rather than the classical step-by-step progression through discrete cell states, HSCs undergo a continuous progression of lineage commitment toward individual mature blood cell types. In mice, HSCs first give rise to MPPs with no or little lineage commitment (MPP1, 5), and then lineage-biased MPPs, such as erythro-megakaryocyte-biased MPP2, myeloid-biased MPP3 and lymphoid-biased MPP4. These lineage-biased MPP subsets preferentially make further commitment to the corresponding lineage but can be instructed to commit towards others by external cues (e.g., inflammation). Surface markers that can be used to identify each murine HSPC and committed progenitor subset are also shown. It is noteworthy that in the revised model, a phenotypically defined progenitor subset (e.g., CMP) should represent a mixture of cells that are located within a certain range of the continuous differentiation path. CMP, common myeloid progenitors; GMP, granulocyte-monocyte progenitors; MEP, megakaryocyte-erythrocyte progenitors; CLP, common lymphoid progenitors; Meg, megakaryocytes; Plt, platelets; EB, erythroblasts; Ery, erythrocytes; Gr, granulocytes; Mφ, macrophages; DC, dendritic cells; NK, natural killer cells; B, B cells; T, T cells.

**Figure 2 ijms-23-01948-f002:**
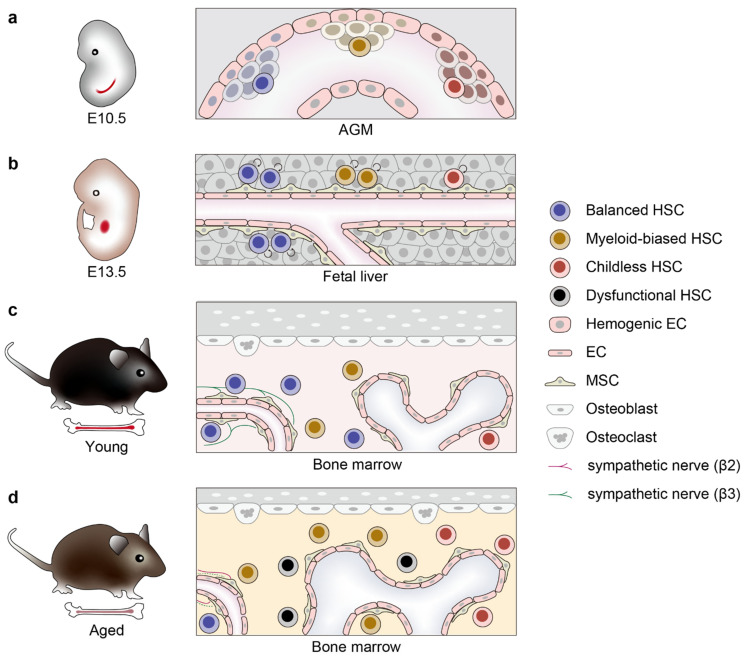
Clonal expansion of functionally distinct HSCs during development and aging. (**a**) Development of HSCs from hemogenic endothelium of the AGM region in mouse embryos at E10.5. HSCs arising from different regions of the AGM can be considered as distinct clones and are shown in different colors. (**b**) At around E13.5, HSCs migrate to fetal liver and clonally expand by self-renewing cell division, while keeping their lineage potential. The degree of expansion differs for each clone [7], and expansion of HSCs with balanced lymphoid and myeloid potential could possibly be favored by the fetal liver niche [52]. (**c**) After birth, HSCs further migrate to bone marrow and most of them enter quiescence in periarteriolar (left) or perisinusoidal (right) HSC niche. (**d**) During aging, HSCs are exposed to various cell-intrinsic and -extrinsic stressors that selectively expand myeloid-biased HSC clones and also induce functionally defective HSC clones, leading to myeloid-biased hematopoiesis and impaired capacity of hematopoietic regeneration. EC, endothelial cells; MSC, mesenchymal stromal cells.

**Figure 3 ijms-23-01948-f003:**
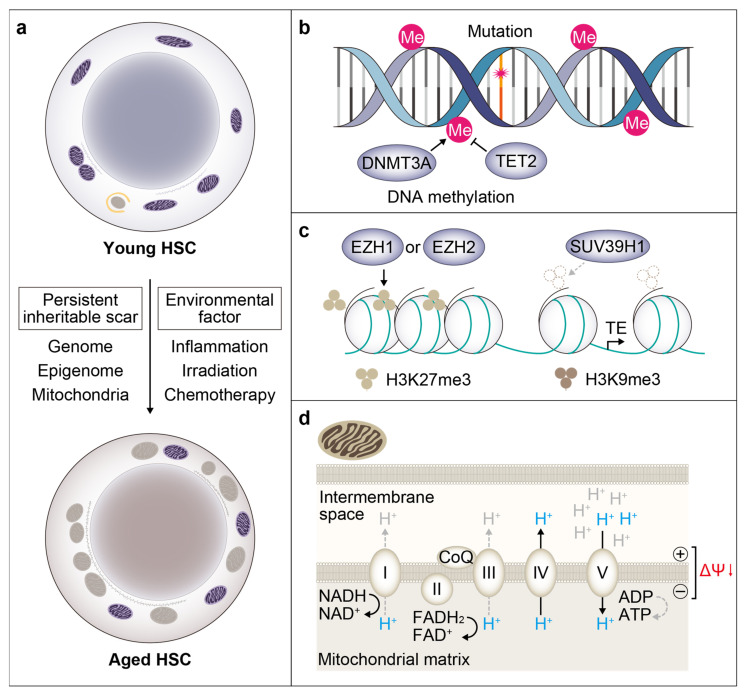
Age-dependent induction of functionally defective HSC clones. (**a**) Factors that are involved in age-associated emergence of HSC clones with impaired function. Intrinsic changes that are inherited by daughter cells upon cell division (e.g., genomic, epigenomic and mitochondrial changes) can alter the property of HSCs and their progeny, thereby increasing the heterogeneity of the HSC pool. Environmental stress, such as exposure to inflammation, irradiation and cytotoxic drugs can cause such inheritable cellular scars in HSCs. (**b**) Inheritable molecular scars on DNA. Mutation and DNA methylation status are copied to synthesized strands and inherited by daughter cells. DNMT3A and TET2, which are frequently mutated in clonal hematopoiesis and hematological malignancies, are involved in DNA methylation and demethylation, respectively. (**c**) Inheritable histone modifications known to be altered in HSCs during aging. Aging alters the activity of EZH1 and EZH2 in HSCs, and changes the genomic regions that are repressed by H3K27me3. The heterochromatin-associated repressive histone mark H3K9me3 decreases with age, which could account for the derepression of transposable elements, such as LINEs, SINEs and LTRs. (**d**) Inheritance of damaged mitochondria. During aging, HSCs are shown to accumulate dysfunctional mitochondria that have low mitochondrial membrane potential (ΔΨ). The damaged mitochondria can be induced and accumulated by various mechanisms, including age-related membrane damage, the impairment of electron transport chain machinery, deregulation in fission and fusion and defective mitophagy.

**Table 1 ijms-23-01948-t001:** Methods used to track HSC clonal behavior in vivo.

Tracking Method	Organism	Labeling Method				Detected Clone Numbers	Ref
		Genotype	Hematopoietic	Off-Target	Efficiency		
Fluorescence (single color)	Mouse	*Scl-Cre^ERT^::Rosa26-LSL-EYFP*	LSK Flk2^−^	-	>90%	1	[36]
			LSK (embryo)	ECs	17%		
		*Tie2-MCM::* *Rosa26-LSL-EYFP*	HSCs	-	1%		[37]
		*Pdzk1ip-Cre^ERT2^::* *Rosa26-LSL-tdTomato*	HSCs	-	30%		[38]
		*Fgd5-ZsGr·Cre^ERT2^:: Rosa26-LSL-tdTomato*	HSCs	BM ECs	10–99%		[39,40,41]
		*KRT18-Cre^ERT2^::* *Rosa26-LSL-EYFP*	HSCs	-	2%		[41]
Fluorescence (multi-color)	Mouse	*Vav1-Cre::Rosa26-Confetti*	All cells	-	~80%	4	[9,45]
		*Mx1-Cre::HUe*	All cells	BM stromal cells, cells in liver, kidney, heart	~80%	15	[43]
	Zebrafish	*drl-Cre^ERT2^::ubi-Zebrabow-M*	Runx1^+^ HSPCs (embryo)	ECs	>95%	20–30	[44,46]
Genome DNA barcode	Mouse	*Rosa26-M2-rtTA::Col1a1-TetO-HSB::Col1a1-CAGGS-Tn-STOP-DSRed*	All cells	All other cells in the body	20–30%	65–905	[47]
		*Rosa26-CreERT2::* *Rosa26-Polylox*	All cells	All other cells in the body	52%	849	[7]
		*Tie2-MCM::* *Rosa26-Polylox*	HSCs (embryo)	-	>95%	382–427	
mRNA barcode	Mouse	*Rosa26-M2-rtTA::tetO-Cas9::Col1a1-cCARLIN*	All cells	All other cells in the body	32–63%	46–321	[48]
		*Tie2-MCM::* *Rosa26-PolyloxExpress*	HSCs (embryo)	-	88%	93–133	[49]
mtDNA mutation	Human	-	All cells	All other cells in the body	>75% (HSCs)	132–315	[50]

LSK, Lin^−^ Sca1^+^ c-Kit^+^; EC, endothelial cells; BM, bone marrow.

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
