# Peer review of "Aging and Clonal Behavior of Hematopoietic Stem Cells"

_ijms, 2022, doi:10.3390/ijms23041948_

Round 1

Reviewer 1 Report

In the current review article, Yamashita and Iwama describe the aspects of hematopoietic stem cells (HSCs) origin, fate, clonal behavior and aging. The article is well written and informative; it contains most of the relevant literature in the field, appropriately cited. It will be an interesting resources for the readers of the journal and for scientists working in the field.
I have some suggestions and integrations on specific points of the review:

Although not a specific focus of the article, I believe that it would be helpful to include a small section on adult HSC differentiation hierarchy showing intermediate progenitors and differentiated output. Maybe also describe their markers. This would help the reader to follow some discussion on specific intermediate progenitors, especially since MPP and other progenitors are cited in several points of the review (for example page 4, rows 135-140). 

In the section on HSC development and clonal expansion, I would add that HSC can be forced to undergo cell division (page 2, row 62) also by traumatic stimulation (for example to compensate massive bleeding) and by different insults to the bone marrow (for example by myeloablative drugs)

Since the authors cite the concept of “childless HSC” from Bowling et al - 2020, it would be good to add a short sentence clarifying the definition.

In the section describing HSC in vivo fate mapping, another study that deserves a citation is Scala et al - Nat Med 2018 “Dynamics of genetically engineered hematopoietic stem and progenitor cells after autologous transplantation in humans”. This study characterized the frequency, dynamics and output of HSPC subtypes by clonal tracking in humans.

The section on the role of inflammation on HSC aging should be expanded, especially since the first author has produced relevant literature on the topic. For example, the effect of TNF-a on aging and malignant hematopoiesis is barely touched on (Yamashita and Passegué - Cell Stem Cell 2019). Other relevant literature on the topic is missing - for example Mann, Metha et al - Cell reports 2018.
When describing mechanisms describing inflammatory factors that influence HSC aging, also cite the role of post-translational modifications (for example, Guillamot et al - Nat Immunol 2019)

In the section describing epigenetic regulators, I would add a small section on the role of ncRNA and miRNA in HSC aging, which is an interesting and emerging concept.

Reviewer 2 Report

Review Yamashita/Iwama

The authors review the aggregate understanding of hematopoietic stem cell behavior during aging. First off, this is an extremely insightful, laborious task they have completed and my comments below are not meant to denigrate it. Striking is that the vast majority of  references is to Nature and Cell papers although I believe it is an undeniable, and universally accepted fact that many of the data, or conclusions there from, are not reproduced. The compelling hypotheses put forth by many of these papers are regularly followed by invariably less prominently published work shedding some doubt on the veracity of the data or hypotheses derived there from. To mention just one example, the label-retaining cells: If indeed these cells divide four times within a mouse’s lifespan, then how is the high labeling efficiency explained which was reported despite the short course of BrdU? One would expect from a review a more critical approach to „paradigms“ and a more inclusive approach to the literature.

Below, I will list several issues which I would invite the authors to consider and in as far as they can concur with the reasoning, to include these in their paper.

The first and most important is: We should probably accept that aging, also aging of thehematopoietic organ, is physiological, not a disease.

Line 11: „lose their function“: They don’t lose their function. They are, in aggregate, possibly less active in certain stress situations, in that they provide less engraftment, etc. But by and large, older people do not have anemia (or if they do, it’s due to iron or vitamin deficiency), they have higher rather than lower neutrophils, and lymphocytes in adults are not made de-novo from stem cells but by proliferation and, to a degree, selection of mature T-cells.

Line 18: The „clonal hematopoiesis“ is DETECTED by virtue of somatic mutations which some HSCs have acquired. So of course CHIP people have somatic mutations, because that’s how we find chip, but it does not mean that mutations cause CHIP or anything of that kind. It is not unlikely that all hematopoiesis is „clonal“ but that we just have no way of identifying the contributing clones because they are not mutated (at least not in the places where we look). Most of these „clones“ do not expand, i.e. the clonal marker is nothing but a marker, is functionally neutral. If a clone expands, this suggests that the mutation conveys the clone a growth advantage but that doesn’t imply that it’s cancer. The co-occurrence of blood cancers or cardiovascular disease with CHIP does not imply that CHIP is a risk factor! CHIP could just as well be a marker of systemic stress – to which different organs respond differently – or maybe a deficiency in damage repair, for instance in DNA damage repair, affecting all organs. The correct reading of the data that are available is that CHIP is more common and the individual clones‘ contribution larger in patients with cardiovascular disease, but the data do not imply a causality in either direction.

Line 39 „can be originated“ – grammar.

Line 43: You have no way of knowing how many clones contribute to a young person’s hematopoiesis. It is very possible that not clone size changes with age, only mutations are accumulated and hence we can distinguish individual clones. Where clones grow over time, the mutations are not only markers, but bestow a growth advantage. Where mutations are non-adaptive, the HSCs will go extinct, and it is not unreasonable to expect this to be the more common event, i.e. mutations in HSCs are much more common than CHIP suggests.

Line 59, „to compensate for their incomplete self-renewal ability“: Quiescent cells are relatively inert to DNA-damaging insults, most importantly ROS. For a potentially immortal cell it is absolutely critical to protect DNA integrity. That, I believe, is why the overwhelming quiescent of the cells at the apex of the hematopoietic hierarchy is so critical. Most of the burden of mature blood cell generation is shouldered not by true stem cells but by early progenitors, already lineage-determined or at least -skewed, with limited self-renewal potential and which are long-lived but not immortal.

Line 61, „lose their self-renewal ability if they are forced to undergo cell division by ex vivo cell culture“: Alternatively: we have just not figured out what we need to provide them to stimulate proliferation without initiating differentiation!

Line 175, „lymphopenia“: Lymphocytes are not replenished from stem cells (much) beyond a certain age - beyond thymic involution. From that point onward, the person lives from the lymphocytes he made earlier, which are long-lived and expand even though they are fully mature.

In other words, I am not convinced by the mantra that the relative lymphopenia in older adults is a consequence of myeloid skewing. Moreover, it seems adaptive to have more innate and less adaptive immunity as we age (and that is why thymic involution is OK to happen). We have „seen“ the most relevant antigens and educated our immune system, built memory, and from then onward new lymphocytes would come at the price of a risk of autoimmunity but would bring little benefit.

And anyway, old people are not lymphopenic, they have a relative abundance of neutrophils and relatively fewer lymphocytes, but not to the degree that the label „lymphopenia“ would stick.

Line 287, „mutation that frequently occur in human (…) do not seem to occur in mouse“: Does this not possibly imply that aging is physiological, and the somatic mutations are bystander events which are egg, not chicken?

Line 380, „irreversible changes in mitochondrial …“: If this were a problem, why would the cells not just make new mitochondria, they have all the tools and all the information, after all? Might this observation imply that the cell has kind of given up and is preparing do die anyway?

Line 455, „Aging process increases the clonality“: I think what the authors mean is "aging reduces the number (or is characterized by a decreasing number?) of active clones contributing to mature hematopoiesis"?

Please give these comments some thought and decide if you want to include some of these arguments in your paper. Overall, the paper is very rich in information, the information is very well presented, but I often miss the executive summary (the authors‘ critical voice) – what, do the authors conclude, is happening when they look at the aggregate data.

Author Response

The authors review the aggregate understanding of hematopoietic stem cell behavior during aging. First off, this is an extremely insightful, laborious task they have completed and my comments below are not meant to denigrate it. Striking is that the vast majority of references is to Nature and Cell papers although I believe it is an undeniable, and universally accepted fact that many of the data, or conclusions there from, are not reproduced. The compelling hypotheses put forth by many of these papers are regularly followed by invariably less prominently published work shedding some doubt on the veracity of the data or hypotheses derived there from. To mention just one example, the label-retaining cells: If indeed these cells divide four times within a mouses lifespan, then how is the high labeling efficiency explained which was reported despite the short course of BrdU? One would expect from a review a more critical approach to „paradigms“ and a more inclusive approach to the literature.

Below, I will list several issues which I would invite the authors to consider and in as far as they can concur with the reasoning, to include these in their paper.

We thank the reviewer for the insightful comments and providing us with the opportunity to consider the critical issues. We have tried to include both supportive and controversial evidence based on the relevance to the topic, selected not solely from high impact journals but also from more specialized journals. As for the controversy of label-retaining cells, we referenced two papers (not from Nature or Cell) that point out the inaccuracy of cell division counting by H2B-GFP (Challen and Goodell. PLoS One 2008, PMID: 18523660, Morcos et al., J Exp Med 2020, PMID: 32302400). Compared to H2B-GFP methods, in vivo administration of BrdU itself is known to promote HSC exit from quiescence and thus achieve relatively high efficiency of labeling the entire HSC pool, which is demonstrated in the referenced paper (Wilson et al., Cell 2008, PMID: 19062086).

The first and most important is: We should probably accept that aging, also aging of the hematopoietic organ, is physiological, not a disease.

We agree that aging is a physiological process and not a disease per se. However, it is indeed one of the most significant risk factors for many diseases, and age-related changes can have a causative effect on such, if not all, diseases. One such example is age-related clonal hematopoiesis, also known as clonal hematopoiesis with intermediate potential (CHIP), and causative link between CHIP and cardiovascular diseases (CVD) has been established at least by several mouse studies (Jaiswal et al., New Eng J Med 2017, PMID 28636844; Fuster et al., Science 2017, PMID 28104796; also reviewed in Jaiswal and Libby. Nat Rev Cardiol 2020, PMID 31406340). Throughout the review, we have made every effort to discriminate between these two things.

Line 11: „lose their function“: They don’t lose their function. They are, in aggregate, possibly less active in certain stress situations, in that they provide less engraftment, etc. But by and large, older people do not have anemia (or if they do, it’s due to iron or vitamin deficiency), they have higher rather than lower neutrophils, and lymphocytes in adults are not made de-novo from stem cells but by proliferation and, to a degree, selection of mature T-cells.

We thank the reviewer for the comment and agree that “HSCs lose their function” is misleading as HSCs do not lose all aspects of their function during aging. We have rephrased it to “HSCs alter their function” (lines 11 and 13).

Line 18: The „clonal hematopoiesis“ is DETECTED by virtue of somatic mutations which some HSCs have acquired. So of course CHIP people have somatic mutations, because that’s how we find chip, but it does not mean that mutations cause CHIP or anything of that kind. It is not unlikely that all hematopoiesis is „clonal“ but that we just have no way of identifying the contributing clones because they are not mutated (at least not in the places where we look). Most of these „clones“ do not expand, i.e. the clonal marker is nothing but a marker, is functionally neutral. If a clone expands, this suggests that the mutation conveys the clone a growth advantage but that doesn’t imply that it’s cancer. The co-occurrence of blood cancers or cardiovascular disease with CHIP does not imply that CHIP is a risk factor! CHIP could just as well be a marker of systemic stress – to which different organs respond differently – or maybe a deficiency in damage repair, for instance in DNA damage repair, affecting all organs. The correct reading of the data that are available is that CHIP is more common and the individual clones‘ contribution larger in patients with cardiovascular disease, but the data do not imply a causality in either direction.

We appreciate the reviewer’s comment and totally agree that hematopoiesis can be clonal even without genetic abnormality, which is indeed the main topic of this review, and that CHIP is detected because it becomes visible due to acquisition of somatic mutations during aging. We intended to use the term “clonal hematopoiesis” to solely indicate “hematopoiesis with mutant HSC clones”, namely CHIP (as defined in lines 308-312), and tried to carefully discriminate it from hematopoiesis governed by HSC clones in general (as depicted in Figure 2 of the revised manuscript). Also, we carefully chose wording to indicate that clonal hematopoiesis is a risk factor for hematological malignancy but not a malignant disease itself. As for causative like between CHIP and hematological malignancy, there are numerous mouse studies that support it such as synergistic roles of CHIP-associated loss-of-function of DNMT3A and TET2 and gain-of-function mutation JAK2V617F in myeloid transformation, as reviewed in Bowman et al., Cell Stem Cell 2018, PMID 29395053. There is also accumulated evidence for a causative link between CHIP and CVD as explained above.

Line 39 „can be originated“ – grammar.

We have corrected the text.

Line 43: You have no way of knowing how many clones contribute to a young person’s hematopoiesis. It is very possible that not clone size changes with age, only mutations are accumulated and hence we can distinguish individual clones. Where clones grow over time, the mutations are not only markers, but bestow a growth advantage. Where mutations are non- adaptive, the HSCs will go extinct, and it is not unreasonable to expect this to be the more common event, i.e. mutations in HSCs are much more common than CHIP suggests.

As referenced in the text (Ganuza et al., Blood 2018, PMID 30782612), the “reduced clonal diversity” in aged hematopoiesis was evidenced by the longitudinal mouse study using Confetti-based in vivo multi-color labelling of hematopoietic cells, not inferred by CHIP in humans. We have included the Confetti-based labelling in Table 1 as it was missing in the original manuscript.

Line 59, „to compensate for their incomplete self-renewal ability“: Quiescent cells are relatively inert to DNA-damaging insults, most importantly ROS. For a potentially immortal cell it is absolutely critical to protect DNA integrity. That, I believe, is why the overwhelming quiescent of the cells at the apex of the hematopoietic hierarchy is so critical. Most of the burden of mature blood cell generation is shouldered not by true stem cells but by early progenitors, already lineage-determined or at least -skewed, with limited self-renewal potential and which are long- lived but not immortal.

We appreciate the reviewer’s comment and interesting hypothesis for the biological relevance of HSC quiescence. However, quiescent HSCs are not “inert” at all to DNA damaging insults. They are rather highly responsive to DNA damaging insults such as irradiation, and indeed susceptible to mutagenesis derived from erroneous repair of DNA double strand breaks via non-homologous end joining (Mohrin et al., Cell Stem Cell 2010). In addition, HSCs are not immortal as they equip intact programmed cell death pathways that can be activated by various stress (Milyavsky et al., Cell Stem Cell 2010, PMID 20619763; Yamashita et al., Cell Stem Cell 2015, PMID 26119234). Instead, we would favor the idea that HSCs are programmed to limit their self-renewal capacity perhaps when they are expanded enough. HSC quiescence allows HSCs to maximize their longevity and minimize the risk of malignant transformation. In other words, when early progenitors provide necessary amount of mature cells to maintain homeostasis, HSCs are reluctant to divide because they want to avoid unnecessary loss due to their incomplete self-renewal ability, but when early progenitors are unable to fulfill the peripheral demand, some of the quiescent HSCs enter the cell cycle to increase the hematopoietic flux (i.e., regeneration). We have added sentences to better explain the biological relevance of HSC quiescence (lines 115-118).

Line 61, „lose their self-renewal ability if they are forced to undergo cell division by ex vivo cell culture“: Alternatively: we have just not figured out what we need to provide them to stimulate proliferation without initiating differentiation!

We agree with the reviewer’s point that HSCs do not necessarily lose self-renewal ability when they undergo cell division but they are just often associated. Thus, we rephrase the text to “often lose their self-renewal ability”.

Line 175, „lymphopenia“: Lymphocytes are not replenished from stem cells (much) beyond a certain age - beyond thymic involution. From that point onward, the person lives from the lymphocytes he made earlier, which are long-lived and expand even though they are fully mature. In other words, I am not convinced by the mantra that the relative lymphopenia in older adults is a consequence of myeloid skewing. Moreover, it seems adaptive to have more innate and less adaptive immunity as we age (and that is why thymic involution is OK to happen). We have „seen“ the most relevant antigens and educated our immune system, built memory, and from then onward new lymphocytes would come at the price of a risk of autoimmunity but would bring little benefit. And anyway, old people are not lymphopenic, they have a relative abundance of neutrophils and relatively fewer lymphocytes, but not to the degree that the label „lymphopenia“ would stick.

We agree that lymphopenia does not always apply for the aged hematopoiesis and thus we decided not to use this word. We also agree that not all of the age-dependent changes in hematopoiesis could solely be attributable to alteration in HSCs and have modified the text to reflect this point (line 259-261).

Line 287, „mutation that frequently occur in human (...) do not seem to occur in mouse“: Does this not possibly imply that aging is physiological, and the somatic mutations are bystander events which are egg, not chicken?

As explained above, we agree that aging is a physiological process and CHIP can be viewed as a type of hematopoiesis governed by HSC clones that become visible due to somatic mutations. We also described in the text that clonal hematopoiesis emerges as a result of age-dependent mechanisms (line 314).

Line 380, „irreversible changes in mitochondrial ...“: If this were a problem, why would the cells not just make new mitochondria, they have all the tools and all the information, after all? Might this observation imply that the cell has kind of given up and is preparing do die anyway?

We thank the reviewer for asking the question. Why quiescent HSCs keep damaged mitochondria remains a major question, but it is suggested that HSCs have relatively low turnover of mitochondria (de Almeida et al., Cell Stem Cell 2017, PMID: 29198942). Thus, impaired activity of DRP1-mediated mitochondrial fission after replicative stress (Hinge et al., Cell Stem Cell 2020 PMID: 32059807), together with inactive basal autophagy upon aging (Ho et al. Nature 2017, PMID: ), might result in accumulation of damaged mitochondria without supply of new mitochondria. Dysfunctional mitochondria can contribute to apoptosis induction, but the accumulated damage in aged HSCs might not be strong enough to induce apoptosis but enough to impair other functions of mitochondria that are responsible for age-related changes in HSC function. We have added a sentence to emphasize the remaining questions as to mitochondria and HSC aging (lines 464-467). In addition, we have modified the text to emphasize that mitochondria damage does not always lead to induction of cell death (lines 407-408).

Line 455, „Aging process increases the clonality“: I think what the authors mean is "aging reduces the number (or is characterized by a decreasing number?) of active clones contributing to mature hematopoiesis"?

We appreciate the reviewer’s question. We agree that “increases the clonality” is misleading and have rephrased the text according to the reviewer’s suggestion (line 553-554).

Please give these comments some thought and decide if you want to include some of these arguments in your paper. Overall, the paper is very rich in information, the information is very well presented, but I often miss the executive summary (the authors‘ critical voice) – what, do the authors conclude, is happening when they look at the aggregate data.

We appreciate the reviewer’s comment and agree that the conclusive sentences were missing at the end of some sections. We have added a few sentences to conclude each section (lines 113-118, 388-391).

Reviewer 3 Report

The authors try to correlate between hematopoietic stem cell (HSC) aging and clonal evolution. They also explained how HSC clones with myeloid skewing and low regenerative potential can be expanded during aging. This is an interesting review and the authors collected vast information from literature and nicely represented it in the manuscript. Collectively, this study was conducted meticulously and appears to be well designed and the collected information supports the conclusion.

Reviewer 4 Report

The review article of Yamashita and Iwama on the clonal behavior of HSCs during aging is very interesting, well structured and exhaustive, even if the topic has already been extensively treated and there are numerous reviews in this regard. Furthermore, the authors' choice to draw attention to the HSC clonal tracking methods makes this review original in its setting. In order to make the reading even easier, I suggest the authors to cite the various panels of figure 2 (a-d) along the text, and to better detail and contextualize the panels b-d.

Round 2

Reviewer 2 Report

This a a very good paper.